# Effective Synthesis of Carbon Hybrid Materials Containing Oligothiophene Dyes

**DOI:** 10.3390/ma12203354

**Published:** 2019-10-15

**Authors:** Piotr Kamedulski, Piotr A. Gauden, Jerzy P. Lukaszewicz, Anna Ilnicka

**Affiliations:** 1Faculty of Chemistry, Nicolaus Copernicus University, Gagarina 7, 87-100 Torun, Poland; gaudi@uni.torun.pl (P.A.G.); jerzy_lukaszewicz@o2.pl (J.P.L.); ailnicka@umk.pl (A.I.); 2Centre for Modern Interdisciplinary Technologies, Nicolaus Copernicus University, Wilenska 4, 87-100 Torun, Poland

**Keywords:** hybrid material, activated carbon, oligothiophene dye, adsorption, porous material, confocal microscopy

## Abstract

This paper shows the first study of the synthesis of hybrid materials consisting of commercial Norit carbons and oligothiophenes. The study presents the influence of surface oxidation on dye deposition as well as changes of pore structure and surface chemistry. The hybrid materials were characterised using Raman spectroscopy, and scanning and transmission electron microscopy (SEM and HR-TEM, respectively). Confocal microscopy was employed to confirm the immobilization of oligomers on the surface of the carbons being investigated. Confocal microscopy measurements were additionally used to indicate whether dye molecules covered the entire surface of the selected commercial Norit samples. Specific surface area and pore structure parameters were determined by low-temperature nitrogen adsorption. Additionally, elemental content and surface chemistry were characterised by means of X-ray photoelectron spectroscopy (XPS) and combustion elemental analysis. Experimental results confirmed that oligothiophene dyes were adsorbed onto the internal part of the investigated pores of the carbon materials. The pores were assumed to have a slit-like shape, a set of 82 local adsorption isotherms was modelled for pores from 0.465 nm to 224 nm. Further, XPS data showed promising qualitative results regarding the surface characteristics and chemical composition of the hybrid materials obtained (sulphur content ranged from 1.40 to 1.45 at%). It was shown that the surface chemistry of activated carbon plays a key role in the dye deposition process. High surface heterogeneity after hydrothermal oxidation did not improve dye adsorption due to specific interactions between surface oxygen moieties and local electric charges in the oligothiophene molecules.

## 1. Introduction

It is now possible to design functional, technologically advanced, and stable materials, which combine the advantages of carbon and organic dyes; this could be pivotal for the development of optoelectronics [1,2]. There is currently great interest in developing new hybrid materials based on various carbon forms, in particular carbon nanotubes [3,4], fullerenes [5,6], graphite [7], graphene [8,9,10], graphene quantum dots [11,12,13], and conducting oligomers or polymers, which potentially can be used in dye-sensitized solar cells (DSSCs) [14,15] or organic light-emitting diode (OLED) displays [16]. In our previous work, we studied the endo- and exohedral hybrid systems of CNT with 2,2’:5’,2”-terthiophene (3T) and α-sexithiophene (6T), both theoretically and experimentally [3]. The analysis revealed that endohedral systems are relatively more stable than their exohedral counterparts. In turn, Martin described the application of different carbon materials in photovoltaics and proved their usefulness for this purpose [17]. The experimental findings prove that such carbon materials are becoming a reality due to their remarkable energy and cost efficiency. An important and less examined phenomenon is the encapsulation of organic dyes inside the pores (mesopores in particular) of commonly available materials, like commercial activated carbons (ACs). Over the last four years, Ranasinghe et al. [18], Joshi [19], and Karki et al. [20] have reported the use of activated carbons of natural origin in the construction of novel counter electrodes (cathodes) of DSSCs. However, the application of carbon materials in an anode design in photovoltaic cells is a real challenge. Therefore, the authors propose a new kind of hybrid material based on commercially available activated carbons and oligothiophene dyes, though certain factors which influence dye deposition on the activated carbons still need to be clarified. Of particular importance is the optimal choice of carbon materials with reference to their structure and surface chemistry (type of functional groups at the surface), with special attention paid to the encapsulation of organic dyes inside the pores of carbon matrixes such e.g., activated carbons. The question here is whether extensive chemical modification of carbon carriers leads to structures of the desired porosity and adsorption ability towards sensitizing dyes, as well as of stability. To this end, two Norit carbons were modified by treating them with 30% concentrated H_2_O_2_ at 50 °C, aiming to increase carbon surface polarity (intensive oxidation), while preserving as much as possible of the original texture properties. Microporous and micro/mesoporous Norit ACs were selected because of their high specific surface area, appropriate size of pores, high purity level, and the fact that they are commercial materials [21,22]. Furthermore, Norit carbons are relatively inexpensive granular ACs, which are often used as adsorbents of diversified species, among others for the decolourization of water and organic solvents.

The current work is built on the hypothesis that it is possible to create hybrid materials (potential anode material for DSSC photovoltaics) using as their basis commercial micro- or mesoporous Norit-type carbons through their modification (controlled adsorption) with oligoheterocyclic molecules. The primary goal of the study was to improve effective dye deposition and increase carbon surface polarity using an oxidative treatment. The surface polarity of AC is mostly determined by the presence of surface moieties such as carboxylic, carbonyl, and hydroxyl functional groups. This and similar concepts were examined in previous studies on sulphur adsorption [23]. Here sulphur compounds were chosen for several reasons: oligothiophenes are small enough to be encapsulated into ACs; they exhibit unique optical and electronic properties; oligothiophene dyes have a well-proven, high chemical and environmental stability. Additionally, properties of the solvent used for the dyes’ dissolution may significantly influence encapsulation. Oligothiophene molecules are still emerging materials for the photonic generation of electricity [24,25,26]. Generally, the dyes are considered to be a low-cost alternative to traditional inorganic semiconductors. This idea has yet to be widely discussed and still has a high level of novelty. It should be noted that the detailed role of surface functionality in organic molecule adsorption is still unknown, as the chemical properties of such compounds are very diverse. Although numerous adsorption studies were performed, there is plenty of room for investigating new AC-dye systems, both theoretically and experimentally. In particular, the role of surface functional groups as supplementary to the micropore filling phenomenon could be ascertained [27]. 

## 2. Materials and Methods

The organic dyes were purchased from Sigma-Aldrich (Polish branch): 2,2’:5’,2”-terthiophene (CAS Number: 1081-34-1) and α-sexithiophene (CAS Number: 88493-55-4). Two types of Norit carbon (Fluka Analytical) were used, i.e., RB3 Norit and PK 1-3 Norit. The dye solution (30 mL) at a concentration of approx. 4 × 10^−3^ M was dissolved in chloroform and added to carbon (0.5–0.6 g) at room conditions. After 7 days (temporary shaking), the carbon was rinsed with chloroform, filtered, and dried at 105 °C for 0.5 h. An analogous procedure was repeated with chemically-modified Norit carbon (10 g of Norit oxidised in 30% H_2_O_2_—70 mL, at 50 °C for 1 h). The hybrid materials obtained with this method are denoted as: Norit_X_Z_yT, where: Norit_X—type of used AC, X = 1—RB3 Norit, X = 2—PK 1-3 Norit. In the case of Z, “ox” suffix means ACs after oxidation. Pristine samples were marked as “raw”. The meaning of _yT is as follows: 3T (2,2’:5’,2”-terthiophene), and 6T (α-sexithiophene). For example, Norit_2_ox_6T denotes PK 1-3 Norit carbon after oxidation and modification by α-sexithiophene. The information on all tested materials is collated in Table 1. 

The hybrid materials were characterised using low-temperature adsorption of nitrogen (specific surface area, pore volumes, and the calculated pore width). The relevant isotherms of all samples were measured at −196 °C using an automatic adsorption instrument, ASAP 2010 (Micromeritics, Norcross, GA, USA). Prior to gas adsorption measurements, the carbon materials were outgassed in a vacuum at 105 °C for 2 h. The low pressure was achieved through the combination of mechanical (ca. p < 1.333 × 10^−3^ hPa) and a turbostatic pump (ca. p < 10^−7^ hPa, HighVac system). For carbon materials, we put samples under the conditions mentioned above in order to remove moisture and solvent. It should be pointed out that exceedingly high temperatures may affect the surface chemistry of our composite materials (causing the reduction/desorption of adsorbed molecules inside the degassing station). 

Differential pore size distribution (PSD) was calculated from the adsorption branch using the Nguyen and Do (ND) method [28,29] with the adsorption stochastic algorithm (ASA) [30]. The pores were assumed to have a slit-like shape, a set of 82 local adsorption isotherms was modelled for pores from 0.465 nm to 224 nm. Micropore volume (V_mi_) was calculated for pores up to 2 nm (based on the cumulative PSD obtained from the ND method). The total pore volume (V_t_) was easily estimated from nitrogen adsorption isotherms at a relative pressure of p/p_s_ ≅ 0.99; V_me_ is the difference between V_t_ and V_mi_. The values of micropore width (x_av_) were calculated using the differential PSD. All measurements discussed below were carried out at ambient conditions. Elemental composition was investigated using a combustion elemental analyser (Vario CHN, ElementarAnalysensysteme GmbH, Langenselbold, Germany). The morphology and microstructure of the materials was analysed by means of electron microscopy (SEM 1430 VP; LEO Electron Microscopy Ltd., Oberkochen, Germany and HRTEM FEI Tecnai F20 X-Twin, Brno, Czech Republic). The carbon samples were dispersed in ethanol and treated with Inter Sonic IS-1K bath for 15 min, then deposited on holey carbon-coated copper grids prior to HRTEM microscopic analysis. Further, SEM/EDX analysis was performed with an energy dispersive X-ray spectrometer (EDX, Quantax 200; detector: XFlash 4010, Bruker AXS, Berlin, Germany). The samples were also analysed using Raman spectroscopy (Renishaw InVia, Renishaw, Laser: Modu-Laser Stellar-REN, Multi-Line-max. Power 150 mW, Leica DM1300M camera Infinity 1; objective: Leica, N PLAN L50x/0.5, Gloucestershire, UK) and X-ray photoelectron spectroscopy (XPS, VG Sci. ESCALAB-210, Fison, Glasgow, UK). XPS measurements were collected using Al Kα radiation (1486.6 eV) from an X-ray source operating at 14.5 kV and 25 mA. Survey spectra were recorded for all samples in the energy range of 0 to 1350 eV with 0.4 eV step. High-resolution spectra were recorded with 0.1 eV step, 100 ms dwell time and 25 eV pass energy. A ninety-degree take-off angle was used in all measurements. Curve fitting was performed using the AVANTAGE software (version 1.62) provided by Thermo Electron, which describes each component of the complex envelope as a Gaussian–Lorentzian sum function; a constant 0.3(±0.05) G/L ratio was used. The background was fitted using the nonlinear Shirley model. Scofield sensitivity factors and the measured transmission function were used for quantification. Additionally, the results were registered with a Leica SP8 confocal microscope (Leica Microsystems GmbH, Wetzlar, Germany) using lasers emitting light at wavelengths of 405 nm (UV lamp). For the Leica confocal microscope, an optimized pinhole x1, long exposure time (400–1000 kHz), 40× or 63× zoom (numerical aperture, 1.3× or 1.4× (HC PL APO CS2)), and diode (405 nm) laser were utilized. The sample was placed, with the addition of glycerine, on a microscope slide with a cover slip. The confocal images were collected sequentially. Leica SP8 (TCS SP8 SMD) software and ImageJ 1.46r software [31] were used for analysis and control experiments. 

## 3. Results and Discussion

The aim of the study was the preparation of new hybrid materials consisting of microporous and micro/mesoporous Norit ACs and their modification by encapsulation of oligothiophene compounds in the studied materials. This goal is consistent with the potential application of these hybrid systems for anode design in DSSC photovoltaics. The investigated Norit ACs are materials of limited surface polarity, which in the case of ACs mainly results from the presence of surface oxygen moieties. The studied oligothiophene dyes are also of a polar species, therefore the authors assumed that intensive oxidation of Norit ACs would increase dipole–dipole interactions between the oxidised carbon surface and the dye molecules. 

### 3.1. Morphology and Structural Characterization

The oligothiophene dyes have a well-proven, high chemical and environmental stability, which could, in combination with different carbon materials, constitute an attractive object of study for applications in various industries, especially electronics. The primary idea was to increase the attractive forces between the dye molecules and the carbon surface by improving carbon surface polarity. Therefore, a number of oxidised Norit AC derivatives were prepared and applied in the dye deposition. Norit_1_raw is a microporous adsorbent (Table 1), as its micropore volume V_mi_ (0.404 cm^3^/g) is 18.5 times bigger than the mesopore volume V_me_ (0.022 cm^3^/g). On the other hand, Norit_2_raw is a micro/mesoporous carbon with a relatively high micropore volume V_mi_ (0.262 cm^3^/g), 1.4 times larger than V_me_ (0.194 cm^3^/g). 

According to the IUPAC classification, all nitrogen adsorption isotherms for the Norit_1 series are type I. Shapes of the isotherms for series Norit_2 confirm the presence of mesopores, since the continuous increase of adsorption can be observed (Figure 1) at higher relative pressure values. All isotherms can be found in the Appendix A. 

The hydrothermal oxidation of both Norits (30% H_2_O_2_ at 50 °C) had only a minor influence on the porosity of these materials (Table 1). After the oxidative treatment, key structural parameters, such as the surface area, micropore volume, and total pore volume (S_BET_, V_mi_, V_t_, and x_av_ respectively), improved slightly in Norit_1, while in Norit_2 they worsened somewhat. Therefore, any difference in the adsorption ability in relation to oligothiophene dyes should be ascribed to “chemical” factors rather than to “porosity/texture” factors. The pore size distribution (PSD) is an important characteristic of porous materials, especially for pore sizes of less than 2 nm. The PSD plots determined for all studied materials are shown in Figure 2. It may be concluded, that a pore width greater than 2 nm does not occur in Norit_1. On the other hand, in Norit_2 the amount of mesopores is greater compared to microporous sorbent, but not significantly so.

A decrease of the pore volume and surface area is a qualitative and quantitative measure of dye encapsulation in the pores (Table 1). Generally, the decrease in micropores’ pore volumes V_mi_ in the Norit_X_yT_ox series did not seem significant when compared to the spectacular decrease in the Norit_X_yT series (non-oxidised Norit carbon), which indicates that dye (3T, 6T) encapsulation took place with intensity, while encapsulation in mesopores was less impressive. The observed decrease in values of V_mi_ and V_t_ for dye-Norit hybrid systems is direct proof that the adsorption of oligothiophenes also takes place in pores, not only on the external surface of the carbon carriers. Surprisingly, the presence of oxygen-containing functional groups in oxidised samples (electron-withdrawing groups) has a negative effect on the adsorption of oligothiophene dyes. We suggest that the decrease is caused by chemical repulsion factors that should not be attributed to steric obstacles for oligothiophene molecule insertion in pores.

Raw Norit carbon has a very high carbon content, i.e., 87.6 wt% for Norit_1_raw and 87.0 wt% for Norit_2_raw samples (Table 1). On the other hand, the modified material shows a decrease in carbon content wt%. The elemental content of carbon is slightly smaller in oxidised Norit carbons than in raw materials, i.e., 81.2 wt% and 86.6 wt% for Norit_1_ox and Norit_2_ox, respectively. Moreover, Norit_1_yT_ox samples, show no decrease in carbon content. However, in Norit_2_yT_ox samples the changes are clearly visible. Qualitatively, these results confirm the adsorption of oligothiophene molecules.

The HRTEM, SEM, and SEM/EDX mapping images of several selected samples, i.e., raw carbon and dye-Norit modified hybrid materials are presented in Figure 3. The images clearly show the amorphous structure and foam-like character of the obtained carbon matrixes. There are visible differences in the structure of the carbon systems before and after dye adsorption, especially for samples from the Norit_2 series (arrow highlights in Figure 3). Based on the SEM images (Figure 3b,e), it can be concluded that both the surface and pore entrances in the ACs are covered by a layer of oligothiophene dyes. In Figure 3c,f, we have confirmation of the presence of sulphur on the surface of hybrid materials. More information, can be found in the Appendix A, including SEM images and SEM/EDX mapping images of raw carbons, oxidised Norit, and dye-Norit modified hybrid materials.

### 3.2. XPS and Raman Spectroscopy Results

X-ray photoelectron spectroscopy was used to further confirm and characterise the dye adsorption on the surface of the hybrid materials. The XPS spectra of Norit_2_6T and Norit_2_ox_6T samples exhibit a chemical composition of sulphur, carbon, oxygen, and a small amount of silicon and nitrogen (Table 2). High-resolution XPS C 1s spectra (Figure 4) exhibit six different carbon peaks at 284.9, 286.5, 287.7, 289.0, 290.3, and 291.6 eV, which can be attributed to the C–C (C-1), C–O– (C-2), C=O (C-3), COOR (C-4), CO (C-5), and π–π transition (C-6), respectively (Table 2). The XPS spectra of dye-modified samples show two peaks at 164.3 and 169.7 eV, which are characteristic of sulphur in thiophene (the former) and certain impurities, mainly in SO_4_^2−^ (the latter) [32,33].

Figure 4e,f shows Raman spectra of the series of hybrid materials which contain α-sexithiophene (6T) excited at a 532 nm laser line. The spectra indicate two main bands at 1335 cm^−1^ (D-band) and 1585 cm^−1^ (G-band). The D-band is associated with structural defects and sp^2^-distortions of partially disordered carbon, while the G-band is common to all sp^2^ carbon forms, which was previously reported by other authors in [34,35,36]. A band at 2700 cm^−1^ called the G’ band is attributed to the overtone of the D band. An additional peak appears at 1441 cm^−1^ (Figure 4e,f), which is not present in the spectra of carbon materials and can be ascribed to the presence of a dye reagent, as was reported in existing literature [4,32].

### 3.3. Confocal Microscopy Results

In order to confirm dye immobilization on the outer surface of commercial Norit carbons, we selected two representative samples, Norit_1_6T and Norit_2_6T and carried out confocal microscopy tests as an alternative qualitative characterization method. Figure 5 presents the confocal microscopy images for the obtained hybrid materials. Dark centres in Figure 5a,b,d–f,h correspond to the presence of carbon material, i.e., commercial Norit carbon. Conversely, bright, colourful areas indicate the presence of 6T dye. In addition, 6T covers the entire surface of the selected commercial Norit carbons. The deposition of a dye on the entire surface of the carbon material is especially evident on an inverted background (arrow highlights in Figure 5c,g). Moreover, this dye deposits much better on the activated carbon surface than in our previous studies on carbon nanotubes [3]. 

## 4. Conclusions

The presented research is a major step towards using prepared and modified commercial carbon materials as an attractive, inexpensive base surface for dye. Norit has a high specific surface area, a properly tailored pore structure, and possible application as a support material for the deposition of oligothiophene dyes. The proposed hybrid materials may potentially be a good system for manufacturing optically sensitive materials for DSSC solar cells [37]. The material can also be used for electrochemical tests, especially potentially for use in ORR/OER electrocatalytic activity [38]. Nitrogen adsorption data, HRTEM, SEM images, and Raman spectra confirm that oligothiophene dyes are adsorbed on the internal part of pores in the investigated carbon materials. Confocal microscopy in particular looks to be a promising technique for indicating the presence of adsorbed 6T molecules on the surface of ACs under investigation. Based on our results, it can generally be concluded that 6T adsorbs much better on the ACs’ surface than the 3T compound. However, despite our expectations, carbon surface oxidation does not increase adsorption of the oligothiophene dye. As a result, oxidised carbon processing cannot be recommended for the production of dye-carbon hybrid materials. 

## Figures and Tables

**Figure 1 materials-12-03354-f001:**
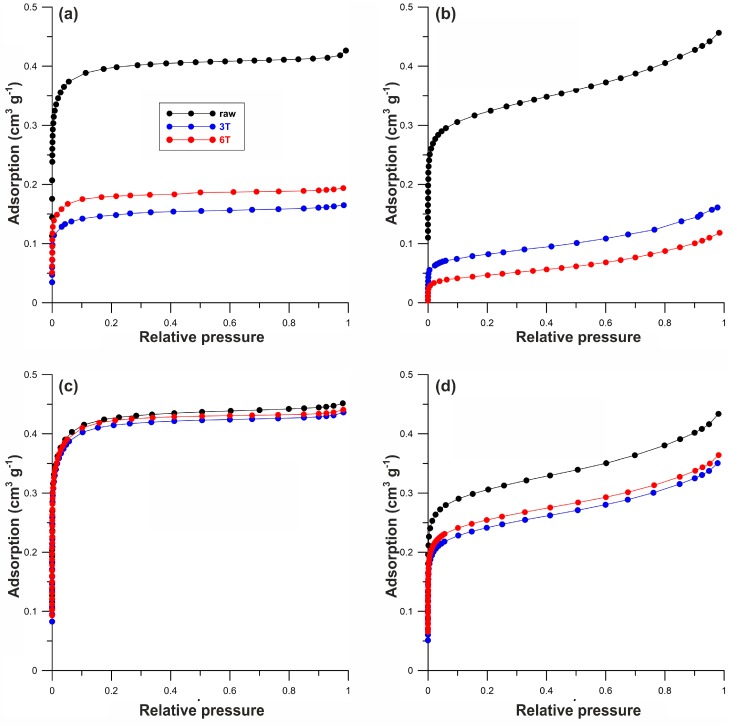
Nitrogen adsorption isotherms of pristine carbon, oxidized, and hybrid materials for (**a**) Norit_1, (**b**) Norit_2, (**c**) Norit_1_ox, and (**d**) Norit_2_ox.

**Figure 2 materials-12-03354-f002:**
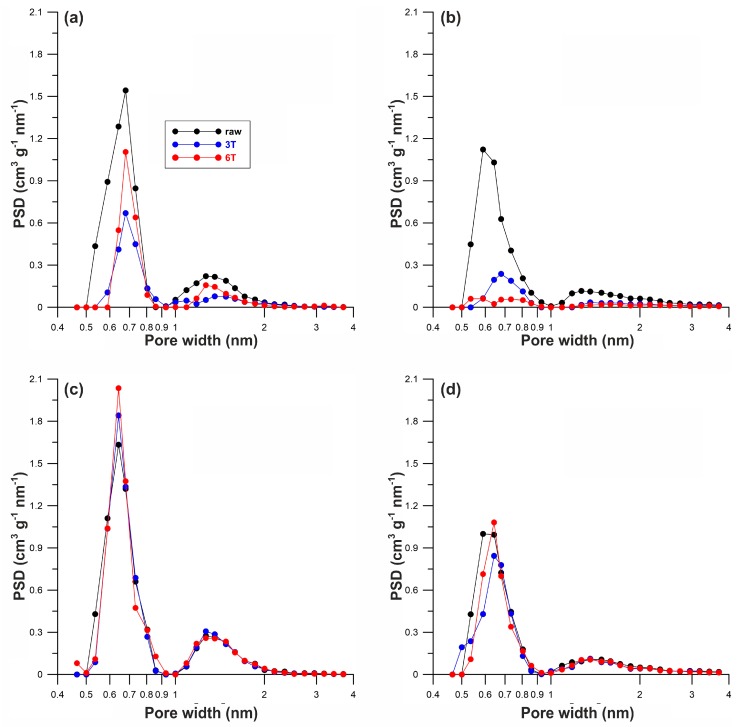
Pore size distribution calculated from the adsorption branch using the Nguyen and Do (ND) method [26,27] for (**a**) Norit_1, (**b**) Norit_2, (**c**) Norit_1_ox, and (**d**) Norit_2_ox.

**Figure 3 materials-12-03354-f003:**
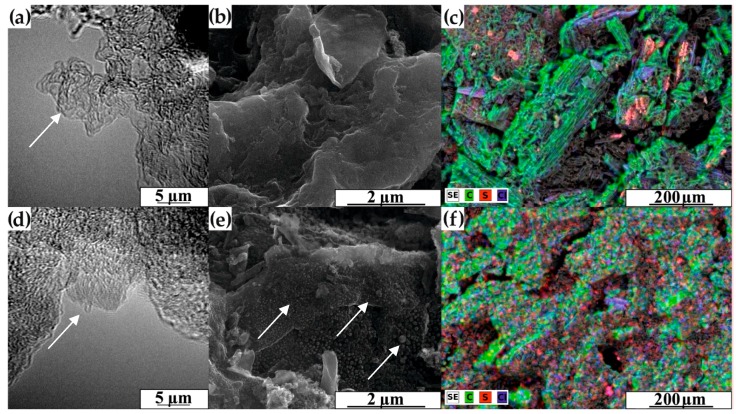
Morphology of the samples: (**a**) HRTEM of Norit_2_raw, (**b**) SEM of Norit_2_raw, (**c**) SEM/EDX mapping of Norit_2_3T, (**d**) HRTEM of Norit_2_3T sample, (**e**) SEM of Norit_2_3T sample and (**f**) SEM/EDX mapping of Norit_2_ox_3T sample; arrow highlights—evidence of dye adsorption.

**Figure 4 materials-12-03354-f004:**
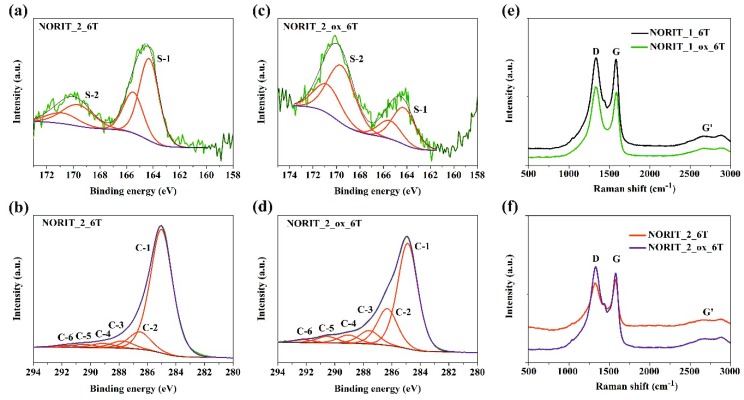
XPS spectra for: (**a**) S2p3 for NORIT_2_6T, (**b**) S2p3 for Norit_2_ox_6T, (**c**) C1s for Norit_2_6T, (**d**) C1s for Norit_2_ox_6T sample, (**e**) and (**f**) Raman spectra of hybrid materials modified with 6T-α-sexithiophene.

**Figure 5 materials-12-03354-f005:**
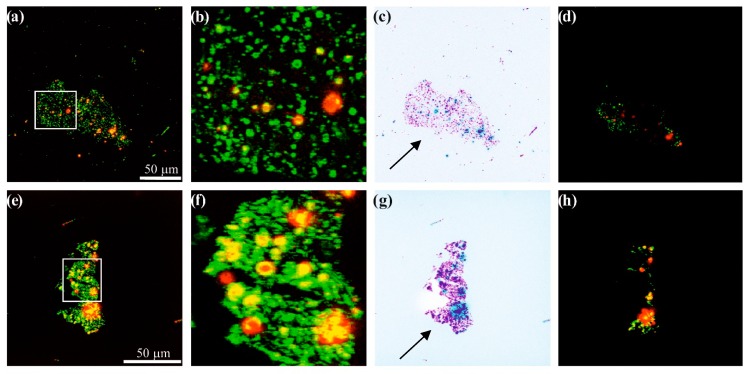
Confocal microscopy images for Norit_1_6T (**a**–**d**) and Norit_2_6T (**e**–**h**) samples, (**b**) and (**f**) present enlarged details from confocal microscopy images shown in (**a**) and (**e**). (**c**) and (**g**) provide inverted-colour versions of the studied materials. (**d**) and (**h**) show the last recorded images after using the laser beam.

**Table 1 materials-12-03354-t001:** Chemical composition and structural parameters of the Norit and hybrid materials.

Sample	Elemental Content (wt%)			S_BET_	V_mi_	V_t_	x_av_
	N	C	H	(m^2^/g)	(cm^3^/g)	(cm^3^/g)	(nm)
Norit_1_raw	1.0	87.6	0.9	855	0.404	0.426	0.861
Norit_1_3T	0.4	77.1	1.5	318	0.162	0.165	0.944
Norit_1_6T	0.4	79.4	1.7	389	0.185	0.194	0.902
Norit_1_ox	0.5	81.2	1.9	902	0.433	0.451	0.857
Norit_1_ox_3T	0.5	81.2	1.1	899	0.420	0.436	0.862
Norit_1_ox_6T	0.5	81.3	1.1	917	0.428	0.441	0.849
Norit_2_raw	0.8	87.0	1.0	705	0.262	0.456	1.140
Norit_2_3T	0.5	85.2	1.7	178	0.089	0.161	1.731
Norit_2_6T	0.4	82.3	1.7	104	0.050	0.118	2.426
Norit_2_ox	0.7	86.6	1.3	642	0.322	0.434	1.137
Norit_2_ox_3T	0.4	77.2	1.0	513	0.253	0.350	1.162
Norit_2_ox_6T	0.5	81.2	1.1	540	0.267	0.364	1.159

**Table 2 materials-12-03354-t002:** Results of the XPS spectra elaboration in C1s, O1, S2p3 region and total content of N and Si of the samples.

Peak	Binding Energy (eV)	Sample		
		Norit_2_raw	Norit_2_6T	Norit_2_ox_6T
		Content (at%)		
C1s (C-1)	285.01	68.73	68.62	45.37
C1s (C-2)	286.54	8.77	9.65	15.81
C1s (C-3)	287.76	4.08	4.00	5.76
C1s (C-4)	289.10	2.47	2.42	3.59
C1s (C-5)	290.47	2.37	1.78	2.77
C1s (C-6)	291.78	1.44	1.09	1.23
Total C	-	87.86	87.56	74.53
S2p3 (S-1)	164.29	-	0.78	0.66
S2p3 (S-2)	169.65	-	0.62	0.79
Total S	-	0.00	1.40	1.45
O1s (O-1)	531.21	3.95	2.78	5.80
O1s (O-2)	533.29	5.23	6.58	14.13
O1s (O-3)	535.22	0.34	0.65	1.06
Total O	-	9.52	10.01	20.99
Total N	-	0.46	0.44	0.64
Total Si	-	2.16	0.58	2.40

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
