# Peer review of "Effective Synthesis of Carbon Hybrid Materials Containing Oligothiophene Dyes"

_materials, 2019, doi:10.3390/ma12203354_

Round 1

Reviewer 1 Report

The manuscript describes a new method to synthesize hybrid material containing Norit carbons and oligothiophenes for DSSC. The authors tried to deposit dyes on commercially available activated carbon surface by adjusting the surface chemical structures. The carbon particles were chemically treated with various chemical agent to functionalize the surface. The authors then carefully examined the surface properties of the treated particles with various characterization tools and the effectiveness of the surface treatments were verified. With the help of surface functional groups, the dye materials were adsorbed on the surface. Overall, the authors presented the methodology well with good figure presentation. Some minor comments are listed below:

The title sounds like a review article and thus is not quite adequate. Please remove the “guidelines,” and change it to “effective synthesis of carbon hybrid materials containing of oligothiophene dyes.”

How much dye is immobilized on the treated carbon? Quantitative measurements should be given.

The dyes might show different optical properties after immobilized on carbon. Any photoluminescence spectra to prove instead of simple confocal test?

Author Response

RESPONSES:

Reviewer 1

The title sounds like a review article and thus is not quite adequate. Please remove the “guidelines,” and change it to “Effective synthesis of carbon hybrid materials containing of oligothiophene dyes.”

Response: We thank the reviewer for this suggestion. We have changed the title to “Effective synthesis of carbon hybrid materials containing oligothiophene dyes” (red color in manuscript).

 How much dye is immobilized on the treated carbon? Quantitative measurements should be given.

Response: We thank the reviewer for this suggestion. For example measurements for two selected series:

                                                          before adsorption      after adsorption

NORIT_2_raw

0.5525

-

NORIT_2_3T

0.5991

0.6848

NORIT_2_6T

0.5630

0.6597

NORIT_2_ox

0.5019

-

NORIT_2_ox_3T

0.5292

0.5799

NORIT_2_ox_6T

0.5018

0.5268

 The dyes might show different optical properties after immobilized on carbon. Any photoluminescence spectra to prove instead of simple confocal test?

Response: We thank the reviewer for this suggestion. We agree with this remark, but have reserved carrying out these tests for the next study and article. In the coming research, we are going to compare the experimental and theoretical optical properties of these materials. Photoluminescence and UV-VIS, in the case of carbon hybrid materials, is useful in determining forbidden energy gap width. That might be helpful in defining the electric/electronic properties of a material, but it is not the main goal of the current study. That is the reason why we did not presently apply it.

Reviewer 2 Report

The authors investigated modified activated carbon that could be utilized as a hybrid material for DSSC photovoltaics. While I didn’t notice any significant issues with the results presented or the scientific approach, the English used (grammar and word choice) made this paper hard to follow. Please seek guidance from native English speaker if possible.

Thank you for the opportunity to review your manuscript. 

Author Response

RESPONSES:

Reviewer 2

The authors investigated modified activated carbon that could be utilized as a hybrid material for DSSC photovoltaics. While I didn’t notice any significant issues with the results presented or the scientific approach, the English used (grammar and word choice) made this paper hard to follow. Please seek guidance from native English speaker if possible. Thank you for the opportunity to review your manuscript.

Response: We thank the reviewer for this suggestion. We have submitted the manuscript for English language correction.

Reviewer 3 Report

Comments regarding the manuscript "Guidelines to effective synthesis of carbon hybrid materials containing of oligothiophene dyes" intended to be published in MDPI Materials. The manuscript presented concerns an interesting and actual subject, the modification of the surfaces of activated carbons

The manuscript appears to be well-written with a logical structure even if the heading might be a little misleading. 

Below some comments on the manuscript

 the heading might be a little misleading, up to me no guidelines are presented, only research results. Authors are suggested to modify the heading

Abstract

The authors could insert some numerical data into the Abstract 

Introduction section

The introduction is rather short and is based on a 25 published references   The aims of the study is explained in a long paragraph but could be described more detailed.   

Materials and methods

The degassing protocol used in the nitrogen measurements appears to use rather low temperatures and short times. This concerns especially the highly micro-porous materials used. Please add some validation results for the protocol. There is an obvious risk that the PSD presented contains errors. The authors also claim that degassing was performed in vacum, please present the pressure used during degassing. Regarding the XPS results, I suggest that table S1is moved into the manuscript after Fig 4 or at line 224. It would clarify the data for the readers.

Conclusions

Authors are suggested to described the possible applications of the products more in detail. There are some mentions about it but it can be more detailed.

Author Response

RESPONSES:

Reviewer 3

The heading might be a little misleading, up to me no guidelines are presented, only research results. Authors are suggested to modify the heading.

Response: We thank the reviewer for this suggestion. We have changed the title to “Effective synthesis of carbon hybrid materials containing oligothiophene dyes” (red color in manuscript).

Abstract. The authors could insert some numerical data into the Abstract.

Response: We thank the reviewer for this suggestion. We have rebuilt the Abstract.

Introduction section. The introduction is rather short and is based on a 25 published references. The aims of the study is explained in a long paragraph but could be described more detailed.

Response: We thank the reviewer for this suggestion. We think that it is not necessary to expand the introduction, but we have added new literature references  (red color in manuscript).

Materials and methods. The degassing protocol used in the nitrogen measurements appears to use rather low temperatures and short times. This concerns especially the highly micro-porous materials used. Please add some validation results for the protocol. There is an obvious risk that the PSD presented contains errors. The authors also claim that degassing was performed in vacum, please present the pressure used during degassing. Regarding the XPS results, I suggest that table S1is moved into the manuscript after Fig 4 or at line 224. It would clarify the data for the readers.

Response: We thank the reviewer for these suggestions. We have added Table S1 as Table 2 to the main manuscript after Fig. 4.

We thank the reviewer for these suggestions. We did not want to go too deep into the details of low-temperature nitrogen adsorption measurements. We decided that they were unnecessary, since ASAP equipments is the most popular apparatus for this type of measurement, and omitted the technical details in the paper. However, we have rebuilt the respective paragraph and added two sentences.

            As we wrote in the article, we made decisions regarding the conditions of temperature and time of outgassing in a vacuum, i.e. 105 °C for 2h. To begin with,, we apologize for the simplification – it is typical jargon used in this context. In our studies, we have used "in a vacuum" to mean that the pressure is used as long as the sample outgasses (partly controlled by software/apparatus, partly by the human operator of the device). Thus, preliminary degassing pressure is equal to ca. 1.333 10-3 hPa. Finally, all samples were outgassed prior to measurements using a turbo-molecular pump (ca. p< 10-7 hPa). In sumary, low pressure was achieved through the combination of a mechanical and turbostatic pump (so-called a HighVac system), with their respective apparatus tests. It should be pointed out that for our samples two hours were enough to reach the equilibrium state.

            It should be noted that the problem of temperature selection is not trivial. For raw/pure carbon materials, some authors have been able to use temperatures up to 200°C or even 250°C most of the times. But this level of heat is not always necessary and is often only justified in the case of narrow micropores which take very long to outgas. This temperature is not even necessary in every case of adsorbents with narrow pores. The same conditions as ours were used previously by other authors for strictly microporous carbon materials, for example, Kruk et al. [M. Kruk, M. Jaroniec, K.P. Gadkaree, Determination of the specific surface area and the pore size of microporous carbons from adsorption potential distributions, Langmuir 145 (1999) 1442-1448] and Kowalczyk et al. [P. Kowalczyk, A.P. Terzyk, P.A. Gauden, R. Leboda, E. Szmechtig-Gauden, G. Rychlicki, Z. Ryu, H. Rong, Estimation of the pore-size distribution function from the nitrogen adsorption isotherm. Comparison of density functional theory and the method of Do and co-workers, Carbon 41 (2003) 1113–1125]. On the other hand, for organic materials, the temperature should be limited to 80°C or even 60°C, depending on the samples (degassing should be done in excellent vacuum for at least one full day). In the case of carbon materials, we put samples under vacuum at 105°C for 2-h to remove moisture and solvent. On the other hand, TGA analysis shows that samples with oligothiophene dyes should be treated carefully upward of 200°C; higher temperatures could be too hot and may affect the surface chemistry of our composite materials (causing reduction inside the degassing station).

The low pressure were achieved through the combination of mechanical pump (ca. p< 1.333 10-3 hPa) and trubostatic pump (ca. p< 10-7 hPa, so-called a HighVac system).

 Conclusions. Authors are suggested to described the possible applications of the products more in detail. There are some mentions about it but it can be more detailed.

Response: We thank the reviewer for this suggestion. A comment has been added to „Conclusions”.

The proposed hybrid materials are potentially a good system for the manufacturing of optically sensitive materials for DSSC solar cells [37]. The material can also be applied in electrochemical tests, with particular potential for use in ORR/OER electrocatalytic activity [38].
